# Concomitant Retinal Alterations in Neuronal Activity and TNFα Pathway Are Detectable during the Pre-Symptomatic Stage in a Mouse Model of Alzheimer’s Disease

**DOI:** 10.3390/cells11101650

**Published:** 2022-05-16

**Authors:** Virginie Dinet, Louiza Arouche-Delaperche, Julie Dégardin, Marie-Christine Naud, Serge Picaud, Slavica Krantic

**Affiliations:** 1Centre de Recherche des Cordeliers, Inserm UMRS 1138, Sorbonne Universités, 75006 Paris, France; virginie.dinet@inserm.fr (V.D.); marie-christine.naud@inserm.fr (M.-C.N.); 2Institut de la Vision, Sorbonne Université, Inserm, CNRS, 75012 Paris, France; louiza.arouche@inserm.fr (L.A.-D.); julie.degardin@inserm.fr (J.D.); serge.picaud@inserm.fr (S.P.); 3Inserm UMRS 938, Team “Immune System and Neuroinflammation”, Hôpital Saint-Antoine, 184 rue du Faubourg St-Antoine, 75012 Paris, France

**Keywords:** retina, Alzheimer’s disease, TNFα signaling, TNFR1, APP/PS1 mouse

## Abstract

The pre-symptomatic stage of Alzheimer’s disease (AD) is associated with increased amyloid-β (Aβ) precursor protein (APP) processing and Aβ accumulation in the retina and hippocampus. Because neuronal dysfunctions are among the earliest AD-related alterations, we asked whether they are already detectable in the retina during the pre-symptomatic stage in a APPswePS1dE9 (APP/PS1) mouse model. The age chosen for the study (3–4 months) corresponds to the pre-symptomatic stage because no retinal Aβ was detected, in spite of the presence of βCTF (the first cleavage product of APP). We observed an increase in ERG amplitudes in APP/PS1 mice in comparison to the controls, which indicated an increased retinal neuron activity. These functional changes coincided with an increased expression of retinal TNFα and its receptors type-1 (TNFR1). Consistently, the IkB expression increased in APP/PS1 mice with a greater proportion of the phosphorylated protein (P-IkB) over total IkB, pointing to the putative involvement of the NFkB pathway. Because TNFα plays a crucial role in the control of neuronal excitability, it is likely that, as in the hippocampus, TNFα signaling via the TNFR1/NFkB pathway may be also involved in early, AD-associated, retinal neuron hyperexcitability. These results further demonstrate the interest of the retina for early disease detection with a potential to assess future therapeutic strategies.

## 1. Introduction

The number of subjects suffering from Alzheimer’s disease (AD), an age-related neurodegenerative pathology, is exponentially increasing world-wide and translating into a global public health problem. However, only around 60% and 10% of cases are diagnosed in high and low-to-middle income countries, respectively (https://www.alzint.org/resource/world-alzheimer-report-2021/, accessed on 21 September 2021). The proportion of affected populations is likely underestimated, due to the late diagnosis and lack of treatment. The AD diagnosis is currently based on scoring clinical symptoms including loss of cognitive functions, such as trouble remembering recent events and eventually total memory loss that interfere with the individuals’ ability to perform daily tasks [1]. Another hallmark of AD, the accumulation of toxic proteins comprising the extracellular deposits of amyloid beta (Aβ) into plaques and intracellular accumulation of abnormally or hyperphosphorylated tau protein (neurofibrillary tangles), is also used for diagnostic purposes in both neuroimaging and biochemical settings.

One of the major concerns about AD is that its diagnosis is currently established years after the initial pathological alterations have occurred [2]. Moreover, the mechanisms linking these initial pathological alterations during the long, clinically silent, pre-symptomatic stage (which spreads over 10–20 years, depending on familial or sporadic etiology) to the diagnosis-relevant clinical symptoms are still largely unknown. The prevailing theory is based on amyloid cascade hypothesis, which posits that overproduction of Aβ from the amyloid precursor protein (APP) initiates a series of events, including synaptic and neuronal network dysfunctions, neuroinflammation with associated microglia and astrocyte activation and hyperphosphorylation of tau, all culminating in widespread neuronal death and neurodegeneration [3].

The increasing body of evidence suggests that transient synaptic hyperexcitability and neuronal network dysfunctions are among the earliest detectable alterations during AD pathogenesis in both animal models and patients [4]. We have previously reported that the mechanisms underlying these early dysfunctions may involve the increase in TNFα during the pre-symptomatic stage of AD [5,6]. Notably, the increased TNFα expression in the hippocampus, the brain region involved in memory formation and among the most vulnerable to AD, coincides with “neuroinflammation-like” phenomena, comprising signs of glia activation in two different mouse models (TgCRND8 [5] and APPswePS1dE9 (APP/PS1) [7]). We have also shown that these early alterations appear concomitant with the altered synchronization of hippocampal neuronal network activities [8,9], in line with the current view on the role of TNFα in the regulation of synaptic excitability and neuronal activity [10].

During the last few years, much attention was been brought to the retina as an accessible part of the brain and a valuable source of putative diagnostic markers for AD [11,12]. For instance, similarities between retinal and cerebral amyloid pathology in rodents and humans have been convincingly demonstrated for the advanced stages of AD [13,14], thus, raising the possibility that initial early alterations may also be similar. In addition, Aβ accumulation begins in the retina prior to the brain, at least in the APP/PS1 mouse model [15]. Furthermore, a proof-of-concept for the translational potential of putative retinal diagnostic markers [14] for human clinical use [16] strongly indicates that the retina may be used to assess the earliest pathological alterations during the pre-symptomatic stage of AD.

Because it is not possible to study pre-symptomatic stage of sporadic AD in humans, we used a well characterized APPswePS1dE9 (APP/PS1) mouse model [17] aged 3–4 months to explore the earliest molecular, structural and functional alterations in the retina. Since the retina has not been studied previously in terms of neuroinflammatory changes in the context of neuronal excitability, we based our age choice on the data previously reported for the hippocampus. Thus, the age of 3–4 months was chosen to study the pre-symptomatic stage, since a previous study reported that, in spite of the absence of amyloid plaques at that age, some “neuroinflammation-like” changes already occur in the hippocampus, yet preceding the cognitive dysfunctions by a few months [18]. These early changes include the TNFα-associated alterations [7] and the hyperexcitability of hippocampal neurons [19]. The age of 9–12 months was chosen to serve as a positive control since it corresponds to (i) the mid-stage of AD-like pathology when virtually all transgenic mice express substantial amounts of cerebral amyloid [17] and (ii) the peak of glial activation when the chronic neuroinflammation is full-blown in the APP/PS1 model [20].

Our data point to a transient hyperexcitability of retinal neuronal response to light that is specifically detectable in dark-adapted conditions (scotopic ERG response) and coincides with triggering the TNFα pathway during the pre-symptomatic stage of AD in the APP/PS1 mouse model.

## 2. Materials and Methods

### 2.1. Animals

Double APP/PS1 transgenic (TG) and their age-matched non-transgenic wild-type (WT) littermate mice aged 3–4 months were used to assess the pre-plaque stage of amyloidosis, corresponding to the pre-symptomatic stage of AD-like pathology [17], whereas older mice (9–12 months) were used as positive controls. The APP/PS1 mice express as a transgene a chimeric DNA sequence, containing the double K595N/M596L Swedish mutation of human APP and human PS1 variant carrying the exon 9 deletion, driven by the mouse prion protein promoter, which directs the expression of the transgene to the central nervous system neurons [17]. The mice were obtained from Jackson Laboratories (Bar Harbor, ME, USA) and bred under standard conditions, which are as follows: 12 h/12 h light/dark cycle (lights on from 7:00 to 19:00) under steady temperature (21 ± 1 °C) and humidity (55 ± 5%), with access to food and water *ad libitum*. Both genders were used in the experiments in a sex-balanced manner.

The animals were treated in accordance with the European Community Council Directive 2010/63/EU for laboratory animal care and the experimental protocol was validated by the Regional Ethical Committee (APAFlS#7609-20l70l1713538844).

The experiments performed in this study were run on two independent cohorts of mice to assess retinal alterations in 3–4 months-old mice, i.e., during the pre-symptomatic stage of AD-like pathology. A total of n = 15 WT and n = 13 TG APP/PS1 mice were used in the first cohort, while the second cohort included n = 5 WT and n = 5 TG APP/PS1 mice. For in vivo experiments (ERG), n = 7 male WT, n = 8 female WT, n = 6 male TG and n =7 female TG were used. Mice from the WT and TG cohorts were used for the group studied at the pre-symptomatic stage (i.e., 3–4 months), whilst in the control group, half (n = 14) of the mice were 9 months old (n = 4 male WT, n = 4 female WT and n = 3 male TG and n =3 female TG), whereas the other half (n = 14) were 12 months old (n = 3 male WT, n = 4 female WT and n = 3 male TG and n = 4 female TG). Mice from the first cohort were used for in vivo ERG and after sacrifice for biochemical experiments (western blot, βCTF ELISA and RT-qPCR), whereas the mice from the second cohort were used for IHC analysis and TNFα ELISA. Because of the limited amount of soluble protein extract that can be obtained from a single retina, it was not possible to run more than 2–3 mini-gels per retina and consequently, the expression of only 2–3 proteins of interest could be assessed by western blot on a single retina. Since in this study we assessed the expression of seven different proteins by western blot, it was necessary to include additional (i.e., which were not submitted to ERG) WT (n = 6) and TG APP/PS1 (n = 11) 3–4 month old mice to the cohort used for western blot experiments. Similarly, additional n = 3 WT and n = 6 TG APP/PS1 mice were required for Aβ1-40 and Aβ1-42 ELISA. Therefore, the total number of mice aged 3–4 months (which were the main focus of this study) was n = 24 and 30 for WT and TG APP/PS1 mice, respectively. For the control group in biochemical and immunohistochemical experiments, mice aged 9 or 12 months were used punctually, as indicated in the relevant figure legends.

The mice of both sexes were used and since no statistically significant difference was observed between the males and females for either genotype, the final comparison between genotypes for each studied parameter was, therefore, carried out using results pooled from both sexes. All the details concerning the total number of mice, the number of males and females used and statistics for the relevant comparisons between the sexes per each studied parameter are given in the Appendix A.

### 2.2. Electroretinography (ERG) and Component Analysis

The ERG was used to assess the function of the photoreceptor pathways in 3–4 and 9–12 month old TG APP/PS1 and WT mice. The mice were dark-adapted overnight (for scotopic recordings) and anesthetized by intraperitoneal injection of a mixture of ketamine 1000 (80 mg/kg, Axience, France) and xylazine (8 mg/kg, Axience, France), under dim red light. The pupils were dilated with tropicamide (Mydriaticum©, Théa, France) and phenylephrin (Neosynephrine, Europhta, Monaco). The cornea was anesthetized with a drop of oxybuprocaine (Théa, France). Body temperature was maintained at 37 °C using a heating pad. The upper and lower lids were retracted to keep eyes open and bulging. ERG responses were recorded with gold-loop electrodes, which were placed through a layer of Lubrithal (Dechra, France) on the corneas of both eyes to measure the summed response with reference to two stainless-steel needle electrodes hooked onto the animals’ cheeks. For the grounding, a needle was subcutaneously inserted in the back of animal (Appendix A). Measurements were performed simultaneously in both eyes using the Ganzfeld VisioSystem device (Siem Biomedicale, France).

Scotopic ERGs were elicited with full-field light flashes corresponding to four levels of stimulus intensity (0.04, 0.32, 3.19, 8 Cd.s/m^2^). The responses were amplified and filtered (1 Hz-low and 300 Hz-high cutoff filters) with a 1 channel DC/AC-amplifier. The stimuli and ERG recordings were coordinated using Visiosystem (Siem Biomedicale, France) program software. The photoreceptor response of the scotopic ERG is represented by the negative a-wave. The positive (scotopic) b-wave reflects the function of the neurons in the inner retina, mostly bipolar cells (Appendix A). ERG (scotopic) measurements were performed with an average of five flash responses from a set of five stimulatory flashes. The responses from each eye of the same experimental group were averaged.

The ERG is generated by contributions from many different retinal cell types, but with appropriate light stimulus, it is possible to selectively stimulate and assess the functional characteristics of a discrete population of retinal neurons. The contribution of rods to vision drops out almost entirely in so-called photopic vision (often designated as “rod-suppressing” condition) because of rods’ response to light saturates. Photopic cone ERGs were recorded in response to a flash (10 Cd.s/m^2^) on a “rod-suppressing” background, i.e., after five minutes of light (25 Cd.s/m^2^) exposure. Each photopic ERG response is the mean of five responses from a set of five stimulatory flashes. Photopic a- and b-waves correspond to the response of n = 15 WT mice (8 females and 7 males) and n = 13 TG mice (7 females and 6 males (Appendix A).

### 2.3. Tissue Processing

For biochemistry, mice were sacrificed by cervical dislocation under light isoflurane anesthesia, retinas were carefully dissected out on ice, snap-frozen in liquid nitrogen and kept at −80 °C until use. The right and left retinas were sampled separately and used for RNA and protein extraction, respectively.

For immunohistochemistry, the whole eyeballs were extracted after cardiac perfusion with 4% paraformaldehyde (Cat n°. 100496, Sigma-Aldrich, ST Quentin Fallavier, France) in a 0.1M phosphate buffer (PBS, Cat n°. P4417-100TAB, Sigma-Aldrich), pH 7.2. The eyeballs were post-fixed for 1 h with the same buffer, then rinsed with PBS, dissected, cryoprotected in sucrose (10 to 30% in 0.1M phosphate buffer, pH 7.2) (Cat n°. 84097, Sigma-Aldrich), frozen and kept at −80 °C until the experiment.

### 2.4. Western Blot

Soluble protein extracts were prepared from each individual retina by homogenization in 10 volumes of lysis buffer (25 mmol/L Tris-HCl, 150 mmol/L NaCl, 1 mmol/L EDTA, pH 7.5), supplemented with protease inhibitor cocktail (Cat n°. 11836 153001, Roche, Merck Millipore, Mannheim, Germany). Lysates were centrifuged at 12,000× *g* for 15 min at 4 °C. The resulting supernatants were collected and the quantity of total protein was determined with the standard BCA (Cat n°. 23235, Thermo Fisher Scientific, Pierce Biotechnology, Rockford, IL, USA) assay. The samples were aliquoted and stored at −80 °C until the western blot and ELISA assays, as described below.

The samples (10 µg proteins of retina lysates) were processed for electrophoresis by adding a Laemli sample buffer and denatured by boiling for 5 min at 100 °C. The proteins were then separated by SDS-PAGE on a Tris-Glycine 4–20% gradient gel (Invitrogen, Thermo Fisher Scientific, Rockford, IL, USA) at 130 V for 2 h. The proteins were transferred to a nitrocellulose membrane (0.45 µm; Amersham, GE Healthcare, Life Sciences, Little Chalfont, UK) at 80 V for 35 min at 4 °C, under agitation. Blots were blocked 1 h at RT with 5% skim milk diluted in Tris-buffered saline (TBS) containing 0.05% Tween (TBS-T) and then incubated with primary antibodies overnight at 4 °C in 1% skin milk, diluted in TBS-T. The nitrocellulose membrane was incubated with the relevant primary antibody over-night at 4 °C (Table 1). After 3 washes of 5 min in TBS-T, the membranes were incubated with secondary antibodies for 1 h at RT in 1% milk TBS-T. The corresponding HRP-conjugated anti-IgG was applied as a secondary antibody (1:2000) in parallel with the HRP-conjugated anti-β-actin IgG (1:2000), used as an internal standard for equal loading. Immunoreactive proteins were revealed using the Western Lightning Chemiluminescence Reagent Plus Kit (Perkin-Elmer, Whatman, Villebon sur Yvette, France), followed by densitometric analysis with Image J software (Win 32, Rasband, WS, Image J, US National Institute of Health, Bethesda, MD, USA). The optical density (OD) measured for each immunoreactive band was normalized to β-actin. The results were expressed as relative optical density (ROD).

### 2.5. ELISA

After eyeball dissection on ice, the retinas were collected and frozen in liquid nitrogen. All the samples were homogenized mechanically to prepare soluble protein extracts, as per procedure used for the western blot.

A commercially available ELISA kit (Cat n°. 27776, IBL International GmbH, Tecan, Männedorf, Switzerland) was used for βCTF (sensitivity 0.02 pmol/L) quantification, according to the manufacturer’s instructions. The range of analysis was between 0.19 and 12 pmol/L. Multiplex Aβ_1-40_; Aβ_1-42_ (Cat n°. K15199E-1, MSD, Meso Scale Diagnostics, Rockville, MD, USA; sensitivity 3.7 and 0.19 pg/mL for AB_1-40_ and AB_1-42_, respectively) was used to quantify human Aβ_1-40_ and Aβ_1-42_ in mouse retinas. ELISA TNFα (Cat n°. 88-7324-22, Thermo Fisher Scientific) was used to quantify retinal TNFα (sensitivity 8 pg/mL).

### 2.6. RT-qPCR

RNA was extracted using TrizolTM (Thermo Fisher Scientific) reagent, according to the manufacturer’s instructions. For RT-PCR analysis, RNAs (300 ng) were reverse-transcribed into cDNA by using oligo-dT primers and SuperScript IV reverse transcriptase (Qiagen, Les Ulis, France), according to the manufacturer’s instructions. Target gene expressions were assessed by real-time qPCR with SYBR Green PCR Master Mix (PE Applied Biosystems, Villebon sur Yvette, France) and selected primers (Table 2) used at the final concentration of 500 nM for target gene amplification in the ABI Prism 7900 sequence analyzer (PE Applied Biosystems, Villebon sur Yvette, France). The amount of the amplification product corresponding to each gene of interest was normalized over an internal standard (HPRT) by calculating the ΔCt (ratio Ct of the target gene over the HPRT). The above equations were applied after the equal amplification efficiencies had been checked, by plotting the amount of cDNA input as a function of ΔCt. The expression was quantified using the 2^−^^ΔΔCt^ method, with the expression of the relevant gene in WT used as a calibrator.

### 2.7. Immunohistochemistry (IHC)

The retinas were fixed with 4% paraformaldehyde (Cat n° 100496, Sigma-Aldrich) in a 0.1M phosphate buffer (Cat n° P4417-100TAB, Sigma-Aldrich), pH 7.2 for 1 h. The right eyes were cryoprotected in sucrose (10 to 30% in 0.1M phosphate buffer, pH 7.2) (Cat n° 84097, Sigma-Aldrich), frozen and kept at −80 °C until the experiment. Transversal 12-μm-thick retina sections were prepared using Leica cryostat (CM3050S). The retinas were incubated in a blocking solution (1% donkey serum (Cat n°. 017-000-121, Jackson immuno-Research, Interchim, Montlucon, France), 2% triton X (Cat n°. X100-500 ML, Sigma-Aldrich), 1% Tween-20 (Cat n°. P7949-100 ML, Sigma- Aldrich) in a phosphate buffer, pH 7.2 at room temperature for 1 h and subsequently over-night at 4 °C with the relevant primary antibodies, as listed in Table 1.

### 2.8. Image Acquisition and Analysis

Immunolabeled tissues were imaged with a Nanozoomer slide scanner (HT-C9600 Hamamatsu) 20× objective or with an Olympus FV1000 laser-scanning confocal microscope with a 20× or 40× objective (UPLSAPO 20XO, NA: 0.85). TNFR1 cell counts were obtained using Imaris software (V.9, Oxford Instruments, Gometz La Ville, France).

### 2.9. Statistical Analysis

GraphPad Prism 5 (GraphPad Software, San Diego, CA, USA) was used for all the statistical analyses. The results are expressed as means ± mean standard error of the mean (SEM). After normality was confirmed using the Shapiro–Wilks test of normality, two-tailed, unpaired Student’s *t*-tests were used to compare the two groups of mice (WT versus TG APP/PS1). In all cases, significance was noted at *p* < 0.05 (*) and *p* < 0.01 (**).

## 3. Results

### 3.1. βCTF, But Not Aβ, Is Expressed in the Retina of Pre-Symptomatic APP/PS1 Mice

APP and its cleavage products were assessed in retinal protein extracts using 6E10 antibody, which recognizes selectively human APP proteins (the expression of which is directed by exogenously introduced human APP transgene). The western blot analysis pointed, as expected, to the human transgene APP expression only in 3–4 month old TG APP/PS1, but not wild-type (WT) littermates (Figure 1A; relevant quantification: Figure 1A’). By analogy, human APP is expressed in the retinal extracts from 9–12 month old TG APP/PS1, but not in those from their WT counterparts, which were used in these experiments as a positive control (Figure 1A; relevant quantification Figure 1A’’). Similarly, western blot using CT20 antibody, directed against the first cleavage product of human APP, i.e., βCTF, could reveal its presence in the retinal extracts from TG APP/PS1, but not WT mice at 3–4 months of age (Figure 1B,B’, respectively), which was, as in the case of APP, in line with the expression of βCTF in older (9–12 months) TG APP/PS1mice, used as a positive control (Figure 1B,B’’, respectively).

Since the western blot approach did not reveal the presence of Aβ in the retinal extracts (data not shown), we next performed a more sensitive ELISA for both Aβ1-42 and Aβ1-40 isoforms. Although ELISA could not detect the presence of either isoform in the retinal extracts from 3–4 month old APP/PS1 mice (data not shown), Aβ1-42 and Aβ1-40 isoforms were identified in the retinal extracts obtained from older (9–12 months) TG APP/PS1 mice, which were used as a positive control (Figure 1C). Regarding βCTF expression, as in the western blot experiments (Figure 1(B’,B’’)), ELISA could also reveal the presence of βCTF in the retinal extracts from TG APP/PS1 mice aged 3–4 months, in addition to the expression observed in TG APP/PS1 samples from mice aged 9–12 months, which were used as a positive control (Figure 1D). Since the antibody used in βCTF ELISA specifically recognizes the human protein, this cleavage product of APP was virtually undetectable in WT retina samples from mice at both studied ages (Figure 1D). Taken together, these data attest that, at the age of 3–4 months, the retina of TG APP/PS1 mice is at a pre-symptomatic stage of AD-like pathology, which precedes significant Aβ accumulations at the later stage of 9–12 months.

### 3.2. Retina of Pre-Symptomatic APP/PS1 Mice Displays Early Neuronal Hyperexcitability

To measure the neuronal activity at this pre-symptomatic stage in TG APP/PS1 mice, we used the in vivo electroretinogram (ERG), which provides a functional measure of various neuronal cell types. The electrical activity of rod and cone photoreceptors, in response to light stimulation, translates into the ERG a-wave, while other neurons contribute to the ERG b-wave generated by the inner retinal components. In order to selectively assess the rod photoreceptor response to light flash, we performed ERG in scotopic conditions, i.e., after over-night adaptation to dark. When comparing a-wave response in TG APP/PS1 mice aged 3–4 months to the response recorded from WT mice in scotopic conditions, we found that a-wave amplitude was significantly increased under a weak flash intensity (0.32 Cd.s/m^2^) in TG APP/PS1 (Figure 2A,B). However, for the higher flash intensities, no difference was detectable between a-wave amplitude in TG APP/PS1 and WT mice at 3–4 months (Figure 2A,B). By contrast, at the age of 9–12 months (positive control), the rod ERG responses in TG APP/PS1 were higher when compared to WT mice at both low (0.04 Cd.s/m^2^) and high flash intensities (3.19 and 8 Cd.s/m^2^) (Figure 2A,C). Concerning the inner retinal response, as assessed by the b-wave amplitude measurement corresponding mainly to the activity of the bipolar neurons, the difference was not statistically significant at the low flash intensity (Figure 2A,D). By contrast, at 9–12 months, the b-wave amplitude was significantly increased in TG APP/PS1 compared to the WT mice at all flash intensities (Figure 2A,E). Under standard lighting conditions used to assess the cone photoreceptor response, the photopic ERG measurements showed no difference at 3–4 months, but hyperexcitability at 9–12 months in TG APP/PS1 compared to the relevant WT mice (Figure 2F,G). Altogether, these data indicate that rod photoreceptors of TG APP/PS1 present an enhanced response, which is already detectable at the low intensity of the light flash during pre-symptomatic stage in TG APP/PS1 mice, whilst for the higher light intensities, this enhancement is detectable only at the mid-stage of AD-like pathology (i.e., 9–12 months). Furthermore, this response enhancement is also detectable at the level of cone photoreceptors and at the inner retinal neurons (mainly bipolar cells) at the overt stage (9–12 months corresponding to the mid-stage of pathology), but such enhancement was not observed at the pre-symptomatic stage.

### 3.3. Early Signs of Gliosis in the Retina of Pre-Symptomatic APP/PS1 Mice

The early stages of AD-like pathology are, at least in the brain, associated with glia (both microglia and astrocyte) activation [5,7,18], known as gliosis. We, therefore, investigated whether TG APP/PS1 mice display increased expression of the canonical microglia (Iba1) and astrocyte (GFAP) markers. We observed an increase in Iba1 immunoreactivity in the inner retina region of TG APP/PS1 as early as at 3 months of age (Figure 3A), pointing to retinal microglia activation. Increased Iba1 immunoreactivity in the inner retina was even more pronounced in the control group, i.e., at the overt stage of pathology in the retina of TG APP/PS1 mice aged 12 months (Figure 3A). Similarly, GFAP-immunoreactivity increased in the inner retina, mostly in the ganglion cells layer of TG APP/PS1 mice at the pre-symptomatic stage (3 months), pointing to astrocyte activation. At the overt stage of pathology, GFAP-immunopositive astrocytes and Iba1-immunopositive microglia were present not only in the plexiform layers, but had also infiltrated the inner and outer nuclear layers (Figure 3A: TG 12 months). By contrast, in the retina of WT mice, both Iba1- and GFAP-immunoreactivities were weak at the two studied ages.

To assess whether the increased expression of Iba1 and GFAP proteins detected at the pre-symptomatic stage is related to a genuine in situ induction of Iba1 and GFAP, we investigated the mRNA expression of these glial markers by qPCR. Both *iba1* and *gfap* mRNAs were significantly more expressed in the retinas of TG APP/PS1 mice than in WT retinas from mice aged 3–4 months (Iba1 WT = 0.99 ± 0.1 TG = 1.8 ± 0.12; *p* = 0.009; GFAP: WT = 1 ± 0.2; TG = 4.2 ± 0.6 *p* = 0.009) (Figure 3B), in agreement with the tissue analysis by immunofluorescence. The observed up-regulation of Iba1 and GFAP expression at both mRNA and protein levels in TG APP/PS1 mice aged 3–4 months suggests an early activation of microglia and astrocytes, which is indicative of gliosis already occurring at the pre-symptomatic AD stage.

### 3.4. Induction of TNFα Signaling Pathway in the Retina of Pre-Symptomatic APP/PS1 Mice

Activated glia produce cytokines, among which some, such as TNFα, are physiologically involved in the control of neuronal excitability [10]. We, therefore, asked whether the observed hyperexcitability, indicated by the increased ERG amplitude and detected gliosis, could be associated with the induction of the TNFα signaling pathway during the pre-symptomatic stage of AD-like pathology. Although *tnf**α* mRNA expression in the retinas of 3–4 month old TG APP/PS1 mice, as compared to WT mice of the same age, was not different (Figure 4A), TNFα protein expression was slightly increased, although remaining below the limit of statistical significance (*p* = 0.0513), as indicated by western blot (Figure 4B). To unequivocally assess whether TNFα level is increased in the retinas of TG APP/PS1 at the pre-symptomatic stage, we performed the more resolutive ELISA assay. Our data point to a significant (more than 40-fold) increase in retinal TNFα protein levels in TG APP/PS1 mice aged 3–4 months (WT: 0.18 ± 0.14 pg/mg protein; TG: 7.48 ± 2.97 pg/mg protein; *p* = 0.03) (Figure 4C). Therefore, TNFα induction occurs at the translational level (rather than transcriptional level), since both western blot and ELISA pointed to the increase in this cytokine expression in TG APP/PS1 *versus* WT mice. However, this difference in protein expression reached significance only in the ELISA experiments, most likely because of the higher sensitivity of ELISA detection.

Because TNFα signaling is mainly mediated by TNFα receptor type-1 (TNFR1) [21], we assessed the putative involvement of these receptors at the pre-symptomatic stage of TG APP/PS1 mice. Both *tnfr1* mRNA and TNFR1protein expressions were up-regulated in the retina of TG APP/PS1 versus WT mice at 3 months (Figure 5A,B). Although at 3 months of age, immunohistochemistry did not confirm a statistically significant (*p* = 0.0571) increase in TNFR1 expression in TG APP/PS1 as compared to WT mice (Figure 5C,D), TNFR1 immunolabelling was detectable in TG APP/PS1 mice, where it was mainly observed in the ganglion cells layer (Figure 5C). Overall, these results indicated an activation of TNFα signaling, due to increased levels of both protein ligand and its receptors expression during the pre-symptomatic stage of AD-like pathology.

### 3.5. TNFα/TNFR1 Induction Correlates with the Phosphorylation of IkBα in the Retina of Pre-Symptomatic APP/PS1 Mice

TNFR1 engagement is a major trigger of inducible NFkB activation [22]. To investigate whether the underlying mechanism by which TNFα exerts its actions in the retina involves NFkB, we focused on the TNFα-mediated activation of NFkB signaling pathway. NFkB is sequestered in the cytoplasm by inhibitory protein IkB and its activation requires the phosphorylation of IkB (P-IkB), which is followed by a degradation of the inhibitor and subsequent translocation of NFkB from the cytoplasm to the nucleus. Activated NFkB acts as a transcription factor and regulates the inflammatory gene expression, including TNFα [22]. We, therefore, asked whether the observed TNFα/TNFR1 up-regulation could be related to NFkB activation through the phosphorylation of IkB. We found that, in spite of similar levels of expression for total NFkB (Figure 6A) and IkB (Figure 6B) between the retinas of TG APP/PS1 and WT mice aged 3–4 months, the level of phosphorylated IkB (Figure 6C), and consequently the ratio phosphorylated-IkB/over total IkB (Figure 6D), increased in the retinas of TG APP/PS1 mice. Since phosphorylated-IkB/total IkB ratio is considered as an NFkB activation index, these results point to the activation of the NFkB pathway in the pre-symptomatic retina of TG APP/PS1 mice. Overall, these data indicate that the triggering of the TNFα signaling pathway in the retina of pre-symptomatic TG APP/PS1 mice could be relayed by down-stream NFkB activation, subsequent to increased phosphorylation of IkB.

## 4. Discussion

The main findings of our study are that, already at the pre-symptomatic stage of AD-like pathology, retinal neurons display hyperexcitablity, as indicated by the increased ERG amplitude. These early alterations of neuronal function occur concomitantly with an increase in βCTF, the first cleavage product of APP, which precedes Aβ generation and deposition. Furthermore, these early alterations are associated with retinal gliosis, an increase in TNFα protein expression and the likely triggering of the down-stream NFkB pathway.

Our study is the first to report an increased photoreceptor response in 3–4-month old TG APP/PS1 mice in comparison to WT controls. The relevant increase in a-wave amplitude was relatively subtle and limited to the low (0.04 and 0.32 Cd.s/m^2^) light stimulus intensities, but it reached statistical significance at 0.32 Cd.s/m^2^. A previous study reported an increase in b-wave amplitude in TG APP/PS1 mice at 3 months of age [23] that was not confirmed in our study, even at 0.04 Cd.s/m^2^ light stimulus intensity (*p* = 0.0724). The only other data concerning functional retinal alterations along the pre-symptomatic stage of AD pathology have been published recently in a 5xFAD model. In this case, the retinal out-put signal measured by pattern ERG decreased as early as at 1 month of age, although a few months prior to cognitive decline in the 5xFAD model [24]. Similarly, another recent study using the same model observed decreased b-wave amplitude in dark-adapted conditions in 3 month old 5xFAD mice, in comparison to the WT controls [25]. However, in contrast to our data in APP/PS1 mice (present study), the previous study did not document a-wave alteration in 3 month old 5xFAD mice [25]. The reasons for these slight differences may stem from different mutations in APP and PS1 expressed in these two models (5xFAD and APP/PS1), as well as from the contribution, or its absence, of tau-related toxicity, in 5xFAD and APP/PS1 mice, respectively. In addition, these retinal alterations of 5xFAD were shown to result from post-receptor abnormalities under dark-adapted conditions. The present report vs. the 5XFAD study shows an enhanced response of rod photoreceptors in APP/PS1, yielding an increase in ERG a-wave amplitude.

In contrast to the early pre-symptomatic stage, retinal dysfunction has been extensively studied at overt mid- and late-symptomatic stages of AD-like pathology in different mouse models. In agreement with our data pointing to a major impairment of both inner and outer neuron function in the retina in control 9–12 month old (mid-stage) APP/PS1 mice, previous studies have reported altered b-wave amplitudes, which were sometimes associated with a-wave amplitude alterations in APP/PS1 mice aged 12 months or older [23,26,27]. Similar alterations have been more recently reported in symptomatic 5xFAD mice [24,28]. Altogether, the agreement of our data obtained for symptomatic (9–12 month old) APP/PS1 mice (used as a positive control) with the previously published data validate the functional experimental approach used here. In addition, our data suggest that during the pre-symptomatic stage, the alteration of retinal functions could be related to rod rather than cone photoreceptors, in contrast to the mid-stage, when both rod and cone photoreceptors are impaired.

Nevertheless, it should be stressed that there is no consensus yet in the literature regarding the AD-related functional alterations in the retina. A previous study that compared 5–6 and 12–13 month old APP/PS1 mice could not detect any difference in either a- or b-wave amplitudes; moreover, it found an amelioration of retinal neuron function in APP/PS1, in comparison to the control WT mice [29]. Of note, by using a western blot approach, this study reported, similarly to our data, the increased βCTF expression at both studied ages without any detectable Aβ, leading the authors to propose that APP processing in the retina fosters the α-amyloidogenic (physiological), over β-amyloidogenic, AD-related pathway [29]. However, in contrast to our approach, the previous study did not use ELISA to assess more accurately (than by western blot) the expression of βCTF. In addition, the retinal Aβ expression has been repeatedly demonstrated in mouse AD models at the overt stage of pathology [24,30,31], including in APP/PS1 mice [15,23,26,27,32,33].

We further found that the above discussed functional impairments, in addition to being coincident with βCTF expression, are also concomitant with the increased expression of microglia and astrocyte markers of activation in the retina of APP/PS1 mice. In agreement with our data, such neuroinflammation-like alterations have been reported previously at the overt stages of AD-related retinal pathogenesis in both APP/PS1 [27] and additional mouse AD-models [24,31,34,35]. By contrast, except our present study, only one previous study has assessed the early, pre-symptomatic stage-associated glia activation in a 3xTgAD model. The latter study reported increased expression of astrocyte marker GFAP by Müller cells, as well as morphological and transcriptomic signs of microglia alterations, thereby, strongly suggesting overall glia activation [35]. These data are in line with our IHC and qPCR data pointing to Iba1 and GFAP induction in the retina of 3–4 month old APP/PS1 mice, as compared to WT controls. Of utmost importance, this previous study also reported a 7-fold increase in the level of TNFα mRNA expression during the pre-symptomatic stage in the retina of 3xTgAD mice [35]. Although the slight increase in TNFα mRNA and protein expression in our qPCR and western blot experiments was not statistically significant, our ELISA experiments pointed to a prominent increase in TNFα in the retinal soluble protein extracts. These differences between TNFα protein expression in the western blot and ELISA experiments are most likely due to the much higher sensitivity of ELISA, as compared to western blot. Combined with the absence of the *tnfra* transcript induction, altogether these data suggest translational, rather than transcriptional, mechanisms behind the increase in retinal TNFα levels, as documented by ELISA.

Furthermore, the increased TNFα level was accompanied by increased TNFR1 expression as attested by qPCR, western blot and IHC, suggesting the triggering of the retinal TNFα/TNFR1 pathway along the early pre-symptomatic stage in APP/PS1 mice. Such putative activation was further indicated by the increased P-IkB/tot-IkB ratio, indicating that the underlying mechanism likely involves the NFkB pathway, despite the absence of the increased NFkB protein expression. However, future studies aimed at the explicit demonstration of the precise mechanisms behind putative NFkB pathway involvement are clearly needed. Remarkably, a single intravitreal injection of oligomeric Aβ yielded NFkB pathway activation and TNFα mRNA induction, which was further associated with microglia activation in the retina of C57/Bl6 mice [36]. Combined, these findings suggest that, similar to our previous data reported in the hippocampus of pre-symptomatic TgCRND8 mice, initial βCTF accumulation in the absence of significant Aβ production may be responsible for the observed early TNFα burst and microglia activation [5]. In addition, by using the same mouse AD model, we also demonstrated that an early burst of TNFα is causally related to the transient hyperexcitability of the hippocampal neurons detectable during the pre-symptomatic stage [6]. Given the previously reported hippocampal neuron excitability in APP/PS1 mice at the same pre-symptomatic stage as used in our present study [19], and taken together with our previous and current data, it is plausible that pre-symptomatic alterations in the retina of APP/PS1 mice, in relation to TNFα induction, retinal neuron hyperexcitability and glia activation, represent a part of the compensatory mechanism initiated by βCTF accumulation. Although this hypothesis remains to be tested directly in the retina, increasing recent evidence is in line with this hypothesis. Thus, in pre-symptomatic Tg2576 mice, βCTF has been directly correlated with entorhinal cortical neuron hyperexcitability [37]. Furthermore, using an elegant transgenic mouse model in which βCTF accumulates in the absence of Aβ accumulation, the involvement of βCTF in synaptic dysfunction has been explicitly demonstrated [38]. Interestingly, both βCTF and Aβ contribute to learning impairment [38] and βCFT levels are significantly higher in postmortem brains of AD patients than in age-matched controls [39]. Of note, previous data from cellular and animal models have postulated that this first cleavage product of APP may act as a trigger of neurodegenerative processes [40].

## 5. Conclusions

In conclusion, although the question of whether the AD-like pathological alterations begin in the retina earlier than in the brain [15,41], or whether these alterations occur concomitantly [42] is still open to debate, it is currently clear that, at least in the APP/PS1 model where the pre-symptomatic stage (in contrast to humans) can be investigated, the brain and retinal pathologies are well correlated [43]. The direct implication of these issues, in addition to the previous proof-of-concept [14], is that retina exploration, due to its accessibility to non-invasive functional examination, should be included as a potential source for pre-symptomatic AD-markers mining, as pointed out in a recent recommendation by the Alzheimer Precision Medicine Initiative (APMI) [16]. In line with this, a study published last year that focused on the evaluation of retinal function by ERG in cognitively healthy patients (asymptomatic in spite of Aβ-positive PET scans) showed important retinal ganglion cell dysfunctions, compared to the age-matched cognitively normal control with Aβ-negative PET scans [44]. Such functional pre-symptomatic assessment in subjects with Aβ-positive PET scans may further include structural examinations of the retinal vessel length, perfusion densities and choroid thickness in the macula area, which were all reported to decrease, at least during the advanced stages of AD [45]. In addition, the results obtained here argue for the consideration of the sampling of ocular fluids for future TNFα assessments as putative early AD biomarkers. The retina, with its easy optical and surgical access, could become an ideal tissue to detect pre-symptomatic AD stages and to evaluate the efficacy of therapeutic treatments, using as out-read specific ocular biomarkers, morphological and functional measures.

## Figures and Tables

**Figure 1 cells-11-01650-f001:**
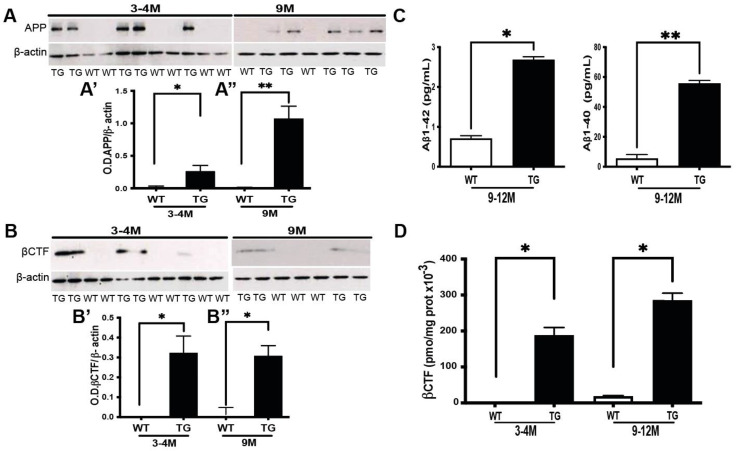
**APP and its cleavage product expression in pre-symptomatic APP/PS1 mouse retina.** (**A**) Representative western blot images of APP expression (apparent Mw = 98 kDa), as revealed with 6E10 antibody directed against the human APP transgene and quantified after normalization over β-actin, used as a loading control. (**A’**) Relative quantification of APP expression (WT n = 6: 3M+3F; TG = 5: 3M+2F) for 3–4 months) and (**A’’**) WT n = 6: 3M+3F); TG = 4: 2M+2F for 9 months). (**B**) Representative western blot images of βCTF expression (apparent Mw = 14 kDa), as revealed with CT20 antibody directed against the cleavage products of human APP. (**B’**) relevant quantification (WT n = 4: 3M+1F; TG = 5: 3M+2F for 3–4 months) and (**B**’’) WT n = 5: 3M+2F; TG = 5: 3M+2F for 9 months). (**C**) Aβ expression was quantified by ELISA against human Aβ_1-40_ (WT n = 3: 2M+1F and TG n = 8 6M+2F *p* = 0.004) with all mice aged 12 months and Aβ_1-42_ proteins (WT n = 3: 2M+1F and TG n = 8: 6M+2F; *p* = 0,01) with all mice aged 12 months. (**D**) ELISA quantification of βCTF (WT n = 4: 2M+2F; TG = 3: 2M+1F for 3–4 months and WT n = 3: 2M+1F; TG = 4: 2M+1F for 9 m and 1F for 12mo). WT = non-transgenic, wild-type; TG = transgenic APP/PS1. * *p* < 0.05; ** *p* < 0.01.

**Figure 2 cells-11-01650-f002:**
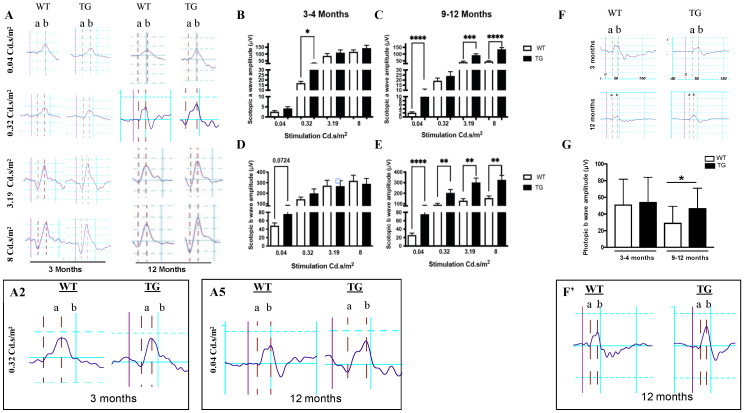
**Functional assessment of the retinal alterations in APP/PS1 mice**. (**A**) Representative a– and b–waveforms of scotopic ERG recorded from 3 and 12 month old WT and TG female mice (F). (A2: enlargement corresponding to the 2nd WT/TG ERG traces example, i.e., 2nd line in the 3 Months row for the indicated light intensity) Scotopic ERG representative data of 3 month old WT and TG F–mice in response to a flash stimulation at 0.32 Cd.s/m^2^. (A5: corresponding to the 5th WT/TG ERG traces example, i.e., 1st line in the 12 Months row for the indicated light intensity) Scotopic ERG representative data of 12 month old WT and TG F–mice in response to a flash stimulation at 0.04 Cd.s/m^2^ (**B**,**C**) WT (white bars) and TG APP/PS1 (black bars) mice ERG, in response to 0.04; 0.32; 3.19 and 8 Cd.s/m^2^ light stimulus intensities corresponding to the rod photoreceptors a–wave amplitude quantification. (**D**,**E**) Inner retinal function assessment by b–wave amplitude quantification. (**F**) Representative of photopic ERG recorded from 3 and 12 month old WT and TG F-mice. (**F’**): Photopic ERG representative recorded of 12 month old WT and TG mice. (**G**) WT (white bars) and TG APP/PS1 (black bars) mice photopic ERG, corresponding to the cone photoreceptors recorded. For scotopic and photopic ERG recordings, a total of n= 15 WT (n= 7 males (M) + 8F) and n = 13 TG APP/PS1 (n= 6M + 7F) were used. Mice from the WT and TG cohort were used at between 3–4 months (n= 7M + 8F WT and n = 6M + 7F TG) and at 9 months (n= 4M + 4F WT and n = 3M + 3F TG) or 12 months (n= 3M + 4F WT and n= 3M + 4F TG). Results are expressed as mean +/− SEM. WT= non-transgenic, wild-type; TG= transgenic APP/PS1. * *p* < 0.05, ** *p* < 0.005, *** *p* < 0.001, **** *p* < 0.0001.

**Figure 3 cells-11-01650-f003:**
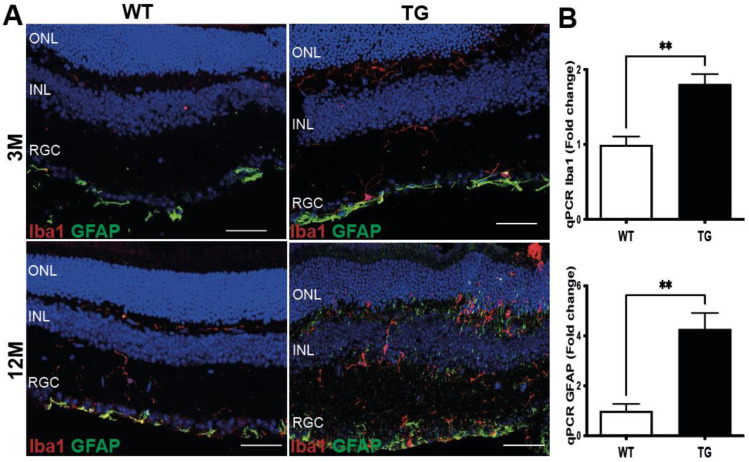
**Gliosis in the retina of pre-symptomatic APP/PS1 mice.** (**A**) Representative images of Iba1 and GFAP immunolabelling in the retina of TG APP/PS1 and WT mice aged 3 (upper panels) and 12 (lower panels) months. Scale bar = 50 μm. (**B**) qPCR analysis of mRNA expression of *iba1* and *gfap* in the retina of TG APP/PS1 and WT mice at 3 months. Results expressed as mean +/− SEM. *** p* < 0.01. qPCR quantification of both GFAP and Iba1: n = 6:3M + 3F for WT mice and n = 4:2M + 2F for TG mice at 3 months.

**Figure 4 cells-11-01650-f004:**
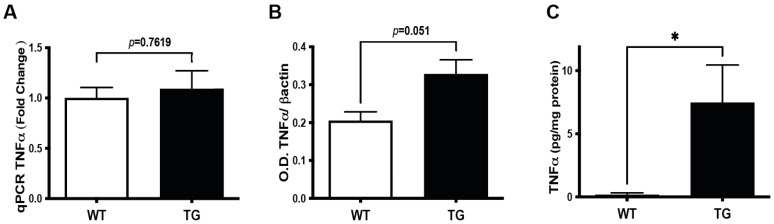
**TNF**α **expression in pre-symptomatic APP/PS1 retina.** (**A**) Transcriptional analysis by qPCR (WT n = 4:2F + 2M; TG n = 6:3F + 3M). (**B**) Analysis of protein expression by western blot with quantification performed after normalization over β-actin, used as a loading control (WT n = 7:3F + 4M; TG n = 6: 3F + 3M). (**C**) ELISA TNFα (WT n = 5:4F + 1M; TG n = 5:3F + 2M) Student’s *t*-test unpaired non-parametric two tailed (Mann–Whitney) * *p* < 0.05.

**Figure 5 cells-11-01650-f005:**
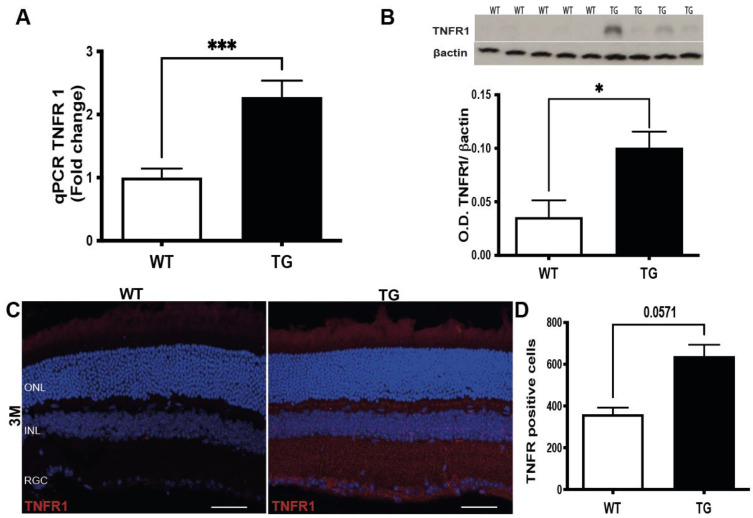
**TNFR1 expression in pre-symptomatic APP/PS1 retina.** (**A**) qPCR analysis of *tnfr1* expression in the retina of TG APP/PS1 versus WT aged 3 months, *p* = 0.0004 (WT n = 6:3F + 3M; TG n = 9:5F + 4M). (**B**) Representative western blot images and relevant quantification of TNFR1 protein expression in the retina of TG APP/PS1 versus WT aged 3–4 months (WT n = 5:2F + 3M; TG n = 7:4F + 3M). (**C**) Representative image of the TNFR1 localization in the retina layers of TG APP/PS1 *versus* WT mice aged 3 months. (**D**) Quantification of TNFR1 positive nuclei in the retina of TG APP/PS1mice at 3 months (WT n = 4:4F; TG n = 3:3M). Immunofluorescence experiments: WT (n = 4) and TG APP/PS1 (n = 3) aged 3 months. Results are expressed as mean +/− SEM. * *p* < 0.05; *** *p* < 0.001.

**Figure 6 cells-11-01650-f006:**
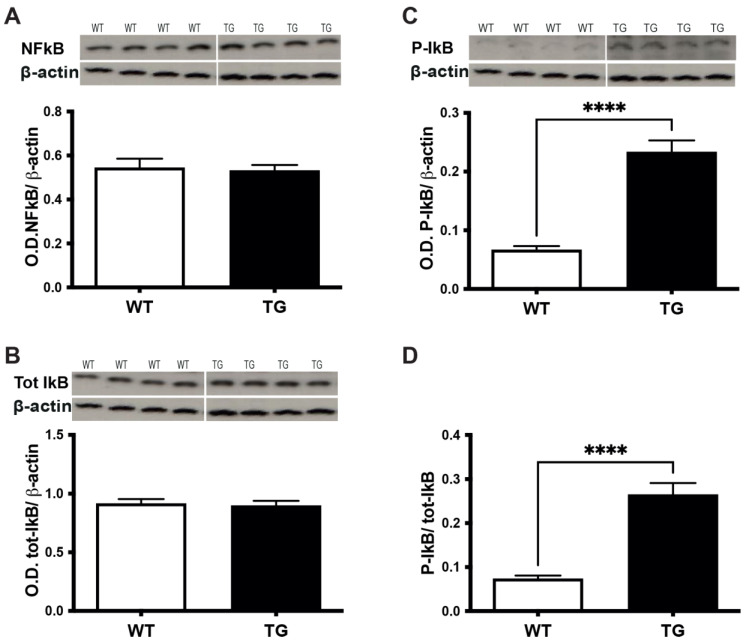
**Assessment of TNFα signaling pathway in pre-symptomatic APP/PS1mouse retina.** (**A**) Analysis of NFkB (WT n = 5:2F + 3M; TG n = 9:5F + 4M), (**B**) total-IkB (WT n = 9:4F + 5M; TG n = 9:4F + 5M) and (**C**) phospho-IkB protein (WT n = 9:4F + 5M; TG n = 9: 4F + 5M) expression by western blot. Representative western blot images are shown above the relevant quantification. All western blot images shown are from the same n = 4 TG and n = 4 WT mice aged 3 months out of a total n = 9 and n = 9 for TG and WT mice, respectively. Quantification was performed after normalization over β-actin, used as a loading control. (**D**) NFkB activation index as expressed by the ratio of phospho-IkB over total-IkB expression. **** *p* < 0.0001.

**Table 1 cells-11-01650-t001:** List of antibodies used to IHC and Western blot experiments.

Antibodies	Ref.	Company	IHC Dilution	Western Blot Dilution
TNFα	AB2148P	Merck Millipore (Fontenay sous Bois, France)	-	1:1000
TNFR1	ab19139	Abcam (Paris, France)	1:500	1:5000
TNFR1	Clone H-5 sc-8436	Santa Cruz Biotechnology, CliniScience, (Nanterre, France)	-	1:2000
AβPP C-T^er^ fragment CT20	171610	Merck Millipore (Fontenay sous Bois, France)	-	1:5000
Aβ 1-16 monoclonal 6E10	SIG-39320	BioLegend (London, UK)	-	1:2000
NFkB p65 (phospho S529)	ab 195838	Abcam (Paris, France)	-	1:1000
IkBα (phospho S32 + S36)	ab 12135	Abcam (Paris, France)	-	1:1000
total IkB	ab 32518	Abcam (Paris, France)	-	1:1000
Iba1	ab5076	Abcam (Paris, France)	1:500	-
GFAP	LS-B4775-50	LSBio (Paris, France)	1:500	-
ß Actin	MBS8533374	CliniScience, (Nanterre, France)	-	1:2000

**Table 2 cells-11-01650-t002:** List of primers used for qPCR experiments.

Gene	Sense 5′-3′ Primer	Anti-Sense 5′-3′ Primer
*HPRT*	TCT AAC TTT AAC TGG AAA GAA TGT C	TCC TTT TCA CCA GCA AGC T
*TNFα*	TCT CAA AAT TCG AGT GAC AAG C	ACT CCA GCT GCT CCT CCA C
*TNFR1*	GAG AAA GTG AGT GCG TCC CT	TGA CAT TTG CAA GCG GAG GA
*Iba1*	CCT GAT TGG AGG TGG ATG TCA C	GGC TCA CGA CTG TTT CTT TTT TCC
*GFAP*	CAG CTG GGC TGT ACA AAC CTT	CAT TGG AAG TGA AGC GTT TCG

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
