# Peer review of "Concomitant Retinal Alterations in Neuronal Activity and TNFα Pathway Are Detectable during the Pre-Symptomatic Stage in a Mouse Model of Alzheimer’s Disease"

_cells, 2022, doi:10.3390/cells11101650_

Round 1

Reviewer 1 Report

Authors' concept regarding "The retina with its easy optical and surgical access could though become an ideal tissue to detect pre-symptomatic AD stages and to evaluate the efficacy of therapeutic treatments" is not a new concept. Authors observed an increase in Iba1 immunoreactivity in the inner retina region of TG APP/PS1 as early as at 3-4 months of age (Figure 3A) pointing to retinal microglia activation. Similarly, GFAP-immunoreactivity is increased in the inner retina, mostly in ganglion cells layer of TG APP/PS1 mice at the pre-symptomatic stage (3-4 months), pointing to astrocyte activation. Inner retinal functions were measured by scotopic ERG in this work. However, the Photopic negative response (PhNR) or pattern ERG has gained interest because they were reported primarily driven by retinal ganglion cells. Because authors already had scotopic ERG data, did the authors try to measure the PhNR during the photopic ERG measurements? 

Author Response

The reviewers’ comments are in italic black, unaltered from the original reviewing reports.

Our responses are in blue. In the revised manuscript (MS), responses to the reviewers are highlighted in yellow, with a comment corresponding to the numbers listed below.

Reviewer1 (R1):

Point 1 (p1) However, the Photopic negative response (PhNR) or pattern ERG has gained interest because they were reported primarily driven by retinal ganglion cells. Because authors already had scotopic ERG data, did the authors try to measure the PhNR during the photopic ERG measurements? 

R1, p1: We would like to thank the reviewer for bringing up this point. Although, we could not record PhNR or pattern ERG, we did perform ERG in photopic conditions, in addition to scotopic ERG even if these photopic data were not shown in the original version of the MS. However, we agree that including the relevant photopic ERG data provide additional, important information concerning the function and integrity of cone photoreceptor cells, which could not be assessed in scotopic (dark-adapted) conditions. The manuscript was revised accordingly to include corresponding method details (page 7, lines 207-216), results (Fig 2 F and G) and their description (page 11-12, lines 361-364; page 12, lines 368-371) to point out that during the pre-symptomatic stage, the alteration of retinal functions come from rod rather than cone photoreceptors, in contrast to the mid-stage (used as a positive control) where both rod and cone photoreceptors are impaired (page 15, lines 490-493).

Reviewer 2 Report

Dr. Dinet and co-authors in their work demonstrate that APP/PS1 mice are characterized by alterations of retina (that means, hyperexcitablity of retinal neurons along with signs of gliosis and neuroinflammation) already at early stages of the pathology (3-4 months of age). These results are of great importance for early diagnosis of Alzheimer's disease (AD) because of easier access of the retina to electrophysiological as well as surgical manipulation compared to the brain tissue.

However, there are several major issues which should be clarify.

  1. The authors pointed that they used male and female mice. It is well known that sex differences may present in AD proceeding. Thus, it is of great importance to point the number of male and female mice; to clarify wheather the presented data were obtain using the only sex or are mix of both sexes and, if so, the data demonstrating the absence of sex differences in studying parameters should be demonstrated.
  2. The authors should clarify, if animals at 9 and 12 months of age represent the same stage of the pathology: the difference in three months may be rather sensible. The authors should point the number of animals at every age.
  3. Line 338. The authors pointed that the difference between TNFα levels according to ELISA was more than 7 times. However, in Figure 4C this difference looks like it was more than 70 times. Besides, the authors should explain such a big difference between results obtained by qPCR and Western blot (WB) analysis and results obtained by ELISA (Fig.4).
  4. Fig.1A,B; Fig.5B; Fig.6A,B,C: the authors should clarify, do the pictures of WB gels represent all samples and, if so, explain the choice of animals' number: 1-4 samples are too little number to make apropriate comparisons between WT and TG mice.

Minor comments:

  1. Line 65: closing parenthesis is absent
  2. The number of animals used (Lines 137-139) should be relocated to Animals subsection; besides, the exact number of male and female mice as well as their exact age (see the major comments 1 and 2) should be clarified.
  3. It is unclear, why the authors used WB and ELISA to analyze the levels of the same protein (βCTF; TNFα).
  4. Protein extraction for WB and ELISA seems the same; thus, Lines 186-190 can be removed.
  5. Lines 276-279: it is unclear, what group of mice (3-4 or 9-12 months of age) this sentence is about.
  6. Fig.2A: the authors should clarify the age of the animals used.
  7. Fig.3B; Fig.4A; Fig.5A: if the number of animals from WT group was more than 1, thus scatters should be presented even if the mean from this group is consider as 100%.

Author Response

Response to Reviewers

The reviewers’ comments are in italic black, unaltered from the original reviewing reports.

Our responses are in blue. In the revised manuscript (MS), responses to the reviewers are highlighted in yellow, with a comment corresponding to the numbers listed below.

Reviewer 2 (R2):

These results are of great importance for early diagnosis of Alzheimer's disease (AD) because of easier access of the retina to electrophysiological as well as surgical manipulation compared to the brain tissue.

We would like to thank the Reviewer for her/his positive feedback.

Major Points

Major Point 1 (MP1) The authors pointed that they used male and female mice. It is well known that sex differences may present in AD proceeding. Thus, it is of great importance to point the number of male and female mice; to clarify wheather the presented data were obtain using the only sex or are mix of both sexes and, if so, the data demonstrating the absence of sex differences in studying parameters should be demonstrated.

R2, MP1: We thank the reviewer for her/his constructive comment since indeed, sex may act as a biological variable even if, at least when considering AD mouse models “sex-based differences in pathological phenotypes are not found in every model and are not consistent even for a single feature across all lines” (Jankosky & Zheng, Mol Neurodegeneration 2017, DOI 10.1186/s13024-017-0231-7). Regarding in particular APP/PS1 mice used in our study, to the best of our knowledge, there is no published data demonstrating the gender differences during the early, pre-symptomatic stage of pathology, which was the main focus of our study. However, considering older mice group which was used here as a positive control, there is a unique previous study in which sex differences were assessed at a single age (i.e. at 9 months). This previous study showed that, at least considering cerebral alterations, female transgenics displayed higher amyloid-beta burden and astrocyte / microglia markers expressions which was accompanied by lower cognitive performance than in male mice (Gallagher et al, Neurodeg Dis 2013, DOI 10.1159/000337458). Of utmost importance however, in both females and males, genotype-driven differences (i.e. differences between transgenic and wild-type mice) were present and qualitatively similar in spite of quantitative differences (Gallagher et al, Neurodeg Dis 2013, DOI 10.1159/000337458). We clearly didn’t find analogous gender differences in terms of neuroinflammatory markers studied neither in the positive control group, aged 9-12 months nor in the studied pre-symptomatic mice group aged 3-4 months (page 6, lines 178-183) (Supplemental Table 1). We would nevertheless like to stress that the mid-stage group (9-12 months-old) was used in the present study only as a positive control (page 5, line 142 where it is first mentioned), since both neuroinflammatory changes and functional ERG alterations have been reported previously in APP/PS1 mice at this mid-stage of pathology. So, data obtained for this age group are not original and the purpose of the relevant experiments performed using this age group was to validate our experimental approach. This fact, i.e. that the pre-symptomatic stage (corresponding to the age of 3-4 months) was the main focus of our study and that older mice were used only as positive control, was already mentioned in the previous version of the MS but is now systematically pointed out whenever it appears in the text of the revised MS. Consequently, the new data obtained in our study are now better distinguished from the already known and previously published data.

Nevertheless, as we fully agree with the Reviewer that gender differences may be of concern we added a statement (page 6, lines 178-183) to underlie the fact that mice of both sexes were used in all experiments and that the data from males and females were pooled since no statistically significant sex-dependent difference was observed for any studied parameter. We also revised the Figure Legends to precise the number of male and female mice used in experiments from which the data shown in each figure were derived. In addition, a paragraph was added in Materials and Methods section (page 5, lines 155-163) to stress that both male and female mice were used while all efforts were done to include the balanced number of mice for each sex. As requested by the Reviewer, the results of relevant statistical analysis for the sex differences for all studied parameter are now shown in Supplemental Tables 1 (for biochemical and immunohistochemical) and Supplemental Tables 2 (for ERG) experiments.

Major Point 2 (MP2) The authors should clarify, if animals at 9 and 12 months of age represent the same stage of the pathology: the difference in three months may be rather sensible. The authors should point the number of animals at every age.

R2, MP2: This point was already partly highlighted in the previous version of the manuscript (page 4, 3rd paragraph: lines 10-12) stating that in this model, the neuroinflammation is full-blown between 8 and 10 months of age (Ruan et al, Curr Alzh Res 2009; 10.2174/156720509790147070). In addition, our choice of age spanning from 9 to 12 months was based on the original characterization of the APP/PS1 model in which the age of 9-12 months was reported to correspond to the mid-stage of AD-like pathology (Fig 2B, 2E and 2H: Jankowsky et al, Hum Mol Genetics 2004, DOI 10.1093/hmg/ddh019). In the particular case of the same strain of APP/PS as the one used in our study (APPswePS1dE9), whereas the mid-stage begins at the age of 9 months (i.e. the age at which all animals express amyloid plaques in the brain), the late-stage of pathology begins by the age of 20 months (Jankowsky et al, Hum Mol Genetics 2004, DOI 10.1093/hmg/ddh019). Consistently, and combining the data from the two aforementioned studies, we considered that in terms of pathology progression, the first third (i.e. 3 months corresponding to the period of 9-12 months) out of 11 months (period from 9 to 20 months spanning the entire mid-stage) would display rather homogenous pathological alterations regarding the neuroinflammatory parameters addressed in our study. Besides, once again, we would like to stress that the mice aged 9-12 months were used as a positive control  since indeed, all data generated in this study concerning this age group have been published previously (please see the references n° 23, 26, 31-37 as per citation in the MS).

In order to keep the main goal of the manuscript straight-forward we preferer not introducing these detailed considerations into the revised MS. However, we added a precision at the end of the Introduction (page 4, lines 127-129) to state that the chosen age of 9-12 months corresponds to the mid-stage of pathology and cited the relevant reference by Jankowsky et al, 2004.

Major Point 3 (MP3) Line 338. The authors pointed that the difference between TNFα levels according to ELISA was more than 7 times. However, in Figure 4C this difference looks like it was more than 70 times. Besides, the authors should explain such a big difference between results obtained by qPCR and Western blot (WB) analysis and results obtained by ELISA (Fig.4).

R2, MP3: In the Results section of the original version of the MS (page 12, line 378) we indeed stated that in ELISA experiments, there was a 7-fold increase in TNFa in TG mice 7.48 +/_2.97 pg/mg protein) as compared to the WT mice where it was only 0.18 +/- 0.14 pg/ mg protein) considering that the cytokine level increases from virtually 0 to 7 pg/mg prot. However, when considering the ratio (7.48/0.18), it gives 41.5-fold increase. We therefore corrected this statement in the manuscript to say that the increase was more than 40-fold (page 13, lines 410-414).  

We moreover added 2 sentences (page 13, lines 414-418) to explain the differences between the results obtained in qPCR, western blot and ELISA experiments:

“Therefore, TNFα induction occurs at the translational (rather than transcriptional level) since both western blot and ELISA pointed to the increase in this cytokine expression in TG APP/PS1 versus WT mice. However, this difference in protein expression reached significance only in ELISA experiments, most likely because of the higher sensitivity of ELISA detection.”

Major Point 4 (MP4) Fig.1A,B; Fig.5B; Fig.6A,B,C: the authors should clarify, do the pictures of WB gels represent all samples and, if so, explain the choice of animals' number: 1-4 samples are too little number to make apropriate comparisons between WT and TG mice.

Thank you for bringing up this point since indeed the figure legends in the original version of the manuscript were not explicit enough. In particular, the gel scans in the original Fig 1A, B contained a small number of samples. These gel scans have been replaced by scans from another experiment with greater number of WT and TG samples (new Fig 1A and 1B). To clarify this issue further, in each relevant figure legend of the revised manuscript, it is now stressed that shown western blot gel scans correspond to the representative images which are only a fraction out of a total number of mice studied in each particular experiment (the total number of mice- males and females- is now clearly stated in Supplemental Table 1). More in detail, from this table, it can be appreciated that a minimum total number of animals used was 4, except for the wild-type mice in the Fig 1 used for Abeta and betaCTF assessment by ELISA which equaled n=3. However, given that no expression of the human APP transgene is expected to be detected by antibodies directed against the human APP protein and its cleavage products (Table 1) in WT animals, we considered that n=3 is sufficient number for the purpose of these experiments. The only other experiment in which the total number of TG mice used was n=3 is the IHC experiment in which the quantification of TNFR1+ cells was performed (Fig 5D). However, we would like to stress that n=3-4 mice per such quantitative IHC experiments is generally acceptable since it usually comprises enumerating 300-600 of cells per section.

Minor Points (mp)/

Minor point 1 (mp1) Line 65: closing parenthesis is absent

R2, mp1: The closing parenthesis was added (page 3, line 102). Thank you for the careful revision!

Minor point 2 (mp2) The number of animals used (Lines 137-139) should be relocated to Animals subsection; besides, the exact number of male and female mice as well as their exact age (see the major comments 1 and 2) should be clarified.

R2, mp2: Agreed and changed accordingly (page 5, lines 155-164) in the revised MS.

Minor point 3 (mp3) It is unclear, why the authors used WB and ELISA to analyze the levels of the same protein (βCTF; TNFα).

R2, mp3: Because the sensitivity of detection is much lower in western blot than in ELISA experiments, in the particular case of betaCTF and TNFalpha we considered that it is important to double check for the relevant protein expression. In addition, for TNFalpha, the relative quantification of the western blot pointed to a difference between WT and TG mice that was at the limit of significance (p=0.0513), which additionally prompted us to double check.

Minor point 4 (mp4) Protein extraction for WB and ELISA seems the same; thus, Lines 186-190 can be removed.

R2, mp4: Agreed and changed accordingly (page 8, lines 254-256) in the revised MS.

Minor point 5 (mp5) Lines 276-279: it is unclear, what group of mice (3-4 or 9-12 months of age) this sentence is about.

R2, mp5: We agree and consequently we completely re-wrote the whole paragraph to present the data obtained in the two studied age groups more clearly (pages 10-11, lines 314-338) in the revised MS.

Minor point 6 (mp6) Fig.2A: the authors should clarify the age of the animals used.

R2, mp6: This point was clarified in the legend of this figure and more in detail in the Supplemental Table 2 which was added to the revised MS (pages 27 and 34-35).

Minor point 7 (mp7) Fig.3B; Fig.4A; Fig.5A: if the number of animals from WT group was more than 1, thus scatters should be presented even if the mean from this group is consider as 100%.

R2, mp7: We agree and amended the MS accordingly.

Reviewer 3 Report

Authors should prepare a major revision for publish in second review.

  1. In figure 1A, could authors check the 6E10 antibody's function in WT mouse?
  2. In figure4, there are no significant differences between mRNA and protein, how do authors explain the changed in ELISA result?
  3. In figure 5B, the TNFR1 protein has low level in all WT mouse, but there are some differences among 4 TG mouse. Tg2 and Tg4  look same level with WT. Could authors explain it?
  4. In figure1a andB, 5B, 6A,B and C. Do authors use the same group WT and TG mouse? If not, could authors describe the animal details and explain why do authors choose the different group mouse? If yes, wyd do authors use different numbers mouse?
  5. Authors observed the p-IKB increased in TG group, but didn't demonstrated the relation between NFKb and IKB in TG.
  6. TNF and NFKB pathways are complex pathway, I don't think only one WB result could demonstrate their relation between NFKb/IKB  and TNFa. Could authors solid their conclusion by adding more evidences?

Author Response

Response to Reviewers

The reviewers’ comments are in italic black, unaltered from the original reviewing reports.

Our responses are in blue. In the revised manuscript (MS), responses to the reviewers are highlighted in yellow, with a comment corresponding to the numbers listed below.

Reviewer 3 (R3):

  1. In figure 1A, could authors check the 6E10 antibody's function in WT mouse?

R3, P1: Yes, 6E10 antibody was used to assess the expression of human APP in both TG and WT mice. As expected, no human APP expression was detectable in WT mice, in contrast to the TG mice (Fig 1A). This is now better explained in the relevant section of the Results, which was entirely reworded (page 10, lanes 314-321).

  1. In figure4, there are no significant differences between mRNA and protein, how do authors explain the changed in ELISA result?

R3, P2: The absence of tnfa mRNA induction in qPCR experiments suggests that the alteration does not occur at the transcriptional level. The difference in the level of TNFa protein expression as assessed by western blot in TG- versus WT mice was at the limit of the statistical significance (p=0.0513). We therefore reasoned that a more resolutive approach such as ELISA could reveal the difference, which indeed turned out to be the case. Combined, these data suggest that the up-regulation of TNFalpha seen in the pre-symptomatic TG mice retina occurs rather at translational than transcriptional level. The relevant issue is discussed more in detail in the revised MS (page 13, lines 414-418) than in its original version.

  1. In figure 5B, the TNFR1 protein has low level in all WT mouse, but there are some differences among 4 TG mouse. Tg2 and Tg4 look same level with WT. Could authors explain it?

R3, P3: At the studied pre-symptomatic stage, TNFR1 expression in the WT mice as assessed by western blot was definitively and invariably close to the background in western blot experiments (Fig 5B), as similarly also observed in WT mice by using IHC approach. (Fig 5C). However, in TG mice, there was an important inter-individual variation such that, for instance, in TG2 and TG4 mice shown in Fig 5B, TNFR1 expression was barely higher than the background whereas it was modest-to-high in other animals (e.g. T3 and T1, respectively) (Fig 5B). We would like to stress that the WB gel shown in Fig 5B (with TG=4 mice) is representative out of a total number of n=7 TG mice studied (this is now outlined in the legend of the Figure 5 and detailed in the Supplemental Table 1).

 Besides, this variability in TNFR1 protein expression was also seen in IHC experiments, the precise reason why we didn’t attempt to quantify the intensity of TNFR1 labeling but rather opted for counting the number of TNFR1-positive cells (Fig 5D).

  1. In figure1a and B, 5B, 6A, B and C. Do authors use the same group WT and TG mouse? If not, could authors describe the animal details and explain why do authors choose the different group mouse? If yes, word do authors use different numbers mouse?

R3, P4: No, it was not possible to perform 7-8 different gels from the soluble protein extracts obtained from a single mouse retina and consequently the representative gel scans shown come from different animals. However, while assessing APP/betaCTF expression on the one hand and NFkB/P-IkB/IkB on the other hand, we limited the assays to the animals for which we had enough soluble protein extracts to perform APP/betaCTF or NFkB/P-IkB/IkB western blots on the same mice. In other words, samples shown on the gel scans in the in Fig 6A-D and Fig 1A and 1B for 3-4 months old mice come from the same animals. Thus, for NFkB/P-IkB/IkB, the western blot was run only on the samples coming from the same animals for which we obtained sufficient amount of soluble protein extracts from each individual animal to run 2 separate gels. These 2 gels were submitted to the electrophoresis in the same migration chamber. After the transfer of proteins onto the nitrocellulose membrane, one gel replicate was blotted with anti-NFkB antibody while the other was immunoblotted with anti-P-IkB antibody. After stripping and provided that ECL revelation confirmed no residual labelling posterior to stripping, the gels were immunoblotted with anti-IkB antibody. This order of procedure was mandatory since anti-P-IkB and anti-IkB antibodies must be applied to the same nitrocellulose membrane in order to be able to compute the P-IkB/IkB ratio. These latter gels were additionally stripped (this was the only case where the gels was stripped twice) for beta-actin assessment. The analogous procedure was applied for APP and betaCTF assessment on 2 gel replicates coming from the same animals and run in the same migration chamber which were subsequently immunoblotted with either 6E10 antibody (for APP expression) or CT20 antibody (for betaCTF expression).

However, sometimes there was simply not enough soluble protein sample to run 2 gel replicates so that, for example, in the 9-month-old mice positive control group in the Fig 1B one (TG3) mouse shown in the Fig 1A is missing and it was replaced by an additional (WT3) sample in the Fig 1B. The reasons for having less soluble proteins from a given retina were often related to the fact that retina dissection is very delicate and sometimes a piece of tissue was lost in the course of dissection.

  1. Authors observed the p-IKB increased in TG group, but didn't demonstrated the relation between NFKb and IKB in TG.

R3, P5: This point has been addressed by a correlative analysis which indicated that when considering only TG mice, the correlation between NFkB and IkB is not significant (r=0.36; p= 0.17). However, when considering the ensemble of WT and TG mice, this correlation did reach significance (r=0.61; p=0.0007), likely because the total number of animals used for the correlative analysis was higher when including both TG and WT mice. However, as correlative analysis cannot serve as a demonstration of causal relationship between NFkB and IkB, we preferred not including these rather speculative considerations to the revised MS.

  1. TNF and NFKB pathways are complex pathway, I don't think only one WB result could demonstrate their relation between NFKb/IKB  and TNFa. Could authors solid their conclusion by adding more evidences?

R3, P6: We agree that one single approach (i.e. WB) is not sufficient to demonstrate the involvement of NFkB. Future studies aimed on this issue are definitively warranted and needed to provide an explicit demonstration and this statement was added in the revised MS (page 17, lines 534-535). Of note, we attached a particular attention in order to not over-discuss our WB data which are presented as an evidence, not as explicit demonstration. However, we would like to stress that the focus of our study was to assess the earliest detectable retinal alterations during the pre-symptomatic stage of AD. In our opinion, this goal was achieved and the amount of work provided by our study corresponds to the current standards of publication.

Round 2

Reviewer 2 Report

The authors took all my comments into account; the manuscript could be accepted in present form.

Reviewer 3 Report

Authors finished all comments' respond.